# Function of Cajal Bodies in Nuclear RNA Retention in *A. thaliana* Leaves Subjected to Hypoxia

**DOI:** 10.3390/ijms23147568

**Published:** 2022-07-08

**Authors:** Sylwia Górka, Dawid Kubiak, Małgorzata Ciesińska, Katarzyna Niedojadło, Jarosław Tyburski, Janusz Niedojadło

**Affiliations:** 1Department of Cellular and Molecular Biology, Nicolaus Copernicus University, Lwowska 1, 87-100 Toruń, Poland; sylwiagorkaa@gmail.com (S.G.); dawidk@doktorant.umk.pl (D.K.); goscies1995@gmail.com (M.C.); karask@umk.pl (K.N.); 2Centre For Modern Interdisciplinary Technologies, Nicolaus Copernicus University, Wileńska 4, 87-100 Toruń, Poland; 3Chair of Plant Physiology and Biotechnology, Nicolaus Copernicus University, Lwowska 1, 87-100 Toruń, Poland; tybr@umk.pl

**Keywords:** Cajal bodies, nuclear retention, intron retention, hypoxia

## Abstract

Retention of RNA in the nucleus precisely regulates the time and rate of translation and controls transcriptional bursts that can generate profound variability in mRNA levels among identical cells in tissues. In this study, we investigated the function of Cajal bodies (CBs) in RNA retention in *A. thaliana* leaf nuclei during hypoxia stress was investigated. It was observed that in *ncb-1* mutants with a complete absence of CBs, the accumulation of poly(A^+^) RNA in the leaf nuclei was lower than that in wt under stress. Moreover, unlike in root cells, CBs store less RNA, and RNA retention in the nuclei is much less intense. Our results reveal that the function of CBs in the accumulation of RNA in nuclei under stress depends on the plant organ. Additionally, in *ncb-1*, retention of introns of mRNA RPB1 (largest subunit of RNA polymerase II) mRNA was observed. However, this isoform is highly accumulated in the nucleus. It thus follows that intron retention in transcripts is more important than CBs for the accumulation of RNA in nuclei. Accumulated mRNAs with introns in the nucleus could escape transcript degradation by NMD (nonsense-mediated mRNA decay). From non-fully spliced mRNAs in *ncb-1* nuclei, whose levels increase during hypoxia, introns are removed during reoxygenation. Then, the mRNA is transferred to the cytoplasm, and the RPB1 protein is translated. Despite the accumulation of isoforms in nuclei with retention of introns in reoxygenation, *ncb-1* coped much worse with long hypoxia, and manifested faster yellowing and shrinkage of leaves.

## 1. Introduction

Frequent occurrence of floods resulting from climate change, including global warming in recent decades, has led to a reduction of 10 to 40% in crop yields [1,2,3]. The 10,000 times slower diffusion and low solubility of oxygen in water lead to hypoxia in floodplain plants [4,5,6]. Some plant species, such as wheat, corn, cotton, tobacco, and rice subjected to hypoxia, have developed morphological adaptations and metabolic mechanisms that increase their tolerance to low oxygen levels [4,6,7,8]. There are two types of defence mechanisms triggered in plants subjected to hypoxia: The first one is the LOES (*Low-O2 Escape Syndrome*) which includes the accelerated growth of aboveground organs, lateral root formation, aerenchyma formation, regulation of stomatal opening and closing, and hyponastic or epinastic growth. The second defence mechanism, called dormancy, consists of waiting out unfavourable conditions by the plant—LOQS *(Low-O2 Quiescence Syndrome*) and involves a slow activation of carbohydrate resources to survive unfavourable conditions with reduced metabolism [6,8,9,10,11,12]. The consequence of hypoxia is the transition from aerobic to anaerobic respiration, the reduction of ATP levels, and the appearance of the so-called “energy crisis” accompanied by the production of anaerobic proteins (ANP), changes in starch metabolism, changes in cytosolic pH and accumulation of reactive oxygen species (ROS) [4,11,13,14,15,16,17]. Plant species-specific metabolic responses have been shown to be associated with the regulation of the expression of genes responsible for hypoxia tolerance at both the transcriptional and post-transcriptional levels [3,18]. In *A. thaliana*, 49 hypoxia response genes (*HRGs*) have been identified [19]. The key plant enzymes of anaerobic metabolism are alcohol dehydrogenase 1 (*ADH1*) and pyruvic decarboxylase (*PDC*) [20,21,22,23]. Almost all expression of HRGs is regulated by ethylene [24]. This gaseous plant hormone is rapidly entrapped in submerged plant tissues and regulates flood-adaptive responses. Then the ethylene receptors located in the ER membrane and Golgi apparatus, i.e., ERS1 (*ethylene response sensor 1*), ERS2, ETR1 (*ethylene resistance 1*), ETR2, and EIN4 (*ethylene insensitive 2 and 4*) are activated. This results in the appearance of the C-terminus of EIN2 in the nucleus, which triggers the transcriptional cascade leading to the expression of group VII ERFs (ethylene response factor) transcription factors which, then, regulate HRGs [25,26,27,28]. In the presence of ethylene, ERFVII (group VII ethylene response factors) is stabilized by impaired proteolysis [24,29].

The mechanisms regulating gene expression in response/adaptation to abiotic stress in plants are not fully understood. At the transcriptional level, phospho-regulation of the CTD domain of RNA polymerase II is an integral part of various biotic and abiotic stress signalling pathways, including microbial-associated molecular patterns/innate immunity signalling [30,31] osmotic stress response [32,33] and hormone signalling [34,35] in *A. thaliana*. Recently, it has been shown that the level of gene expression is influenced not only by the number of transcripts but also by their localization in the cell, for example, the spatial regulation of gene expression by the separation of mRNA transcription and translation by the nuclear envelope. The precise regulation of transcript transport from the nucleus to the cytoplasm prevents the translation of defective or unnecessary mRNAs in the cell [36]. Additionally, it has been shown in mammals that nuclear retention of RNA controls transcriptional bursts that can generate profound variability in mRNA levels among identical cells and in a single cell over time [37]. Moreover, we have demonstrated that nuclear retention is also the mechanism of mRNA storage in the nucleus of root cells of *A. thaliana* during hypoxic stress, and after the removal of stressful conditions, the accumulated transcripts move to the cytoplasm during reoxygenation [38]. The selective storage of mRNA in the cell nucleus, which can be quickly used when adverse conditions end, may be a plant strategy to survive in hypoxic conditions and recover in normal conditions. Recently, Lee and Bailey-Serres (2019) [39] revealed the nuclear transcriptome in hypoxia-treated *A. thaliana.* The analysis of RNA bound to RNA Ser2P polymerase II and nuclear RNA demonstrates that genes encoding ribosomal proteins, metabolic enzymes, and some components of the photosynthetic apparatus are synthesized and retained as partially or completely processed transcripts in the nucleus during hypoxia, a situation reversed by reaeration. However, the mechanism of transcript retention in the nucleus still remains incomprehensible. The accumulated mRNAs lack significant sequences or features that can explain their temporary retention or release from the nucleus [40].

Our recent results indicate the function of Cajal bodies (CBs) in the accumulation of poly(A^+^) RNA in the nucleus during hypoxic stress in the root cells of three species: *A. thaliana*, *L. luteus*, and *A. cepa* [38,41]. In animal cells, CBs are identified by coilin, but in plant cells, the U2B” and U2 snRNA are additional markers [38,42,43,44]. CBs are multifunctional nuclear domains found in the nuclei of plant and animal cells [45,46] that are involved in the maturation of different types of RNA, mainly snRNPs, such as snRNA and snoRNA. Processes involving the role of CBs in snRNP biogenesis include snRNA methylation, scaRNP-mediated pseudouridylation, and the formation of a preinitiation complex consisting of U4, U5, and U6 snRNPs [47,48]. It has also been demonstrated that CBs participate in mRNA storage in both somatic and generative cells [38,49]. The *A. thaliana ncb-1* mutants devoid of CBs show a lower tolerance to hypoxia than the wild type, which might be connected to a much lower accumulation of poly(A^+^) RNA in the nucleus of *ncb-1* in comparison to the wild-type. The presence of active RNA polymerase II and SR proteins has not been found in CBs so far, which suggests that they are not the site of transcription and splicing. This means that CBs can participate in the later stages of the post-transcriptional regulation of gene expression (PTGR) in plant cells exposed to abiotic stress. CBs are associated with the localization of certain types of RNA in the nucleus. However, it is not known how this phenomenon is related to other PTGRs, including alternative splicing, alternative polyadenylation, and interference [50,51]. The analysis of the transcriptome of *A. thaliana* has shown that approximately 40% of the transcripts contain intron sequences. However, these transcripts often contain premature termination codons (PTCs) [52,53,54]. These transcripts should be degraded by NMD (nonsense-mediated mRNA decay), but most of them escape this pathway. Moreover, their significance during stress response is not fully understood.

In this study, we determined the roles of Cajal bodies, in nuclear RNA accumulation and intron retention in the RPB1 mRNA, in leaves of *A. thaliana* subjected to hypoxia stress.

## 2. Results

Mutant *ncb-1* with a complete absence of CBs coped much worse with long hypoxia (5 days), and manifested faster yellowing and shrinkage of leaves compared to wt plants (Appendix A). Some of the plants were reoxygenated (Appendix A). Then, these plants developed inflorescence stems and pods with seeds, indicating that 5-day hypoxia is not lethal for either plant types. The *ncb-1* mutant was obtained by U2B”: GFP line treated with EMS. In *ncb-1* a change from G to A has occurred at the 5’ end of the exon of coilin and resulted, in a lack of CBs in nuclei [54]. First, the biogenesis of ethylene, the hormone that induces plant responses to hypoxia, was investigated by immunolocalization (Appendix A) and quantity of ACC (ethylene precursor) (Appendix A), ACS2 (1-aminocyclopropane-1-carboxylate synthase) and ACO2 (1-aminocyclopropane-1-carboxylic acid oxidase) mRNA measurements (Appendix A) in leaf cells during normoxia, hypoxia, and reoxygenation. ACC occurred mainly in tonoplasts, vacuoles, and chloroplasts in all the studied stages (Appendix A). The localization of the ethylene precursor varied in the stages analysed. Significant changes were not observed during hypoxia between wt and *ncb-1* at up to 5 days of oxygen deprivation (Appendix A). A stronger signal during prolonged hypoxia occurred in the mutant (Appendix A). However, reaeration was accompanied by a change in ACC distribution, and the ethylene precursor was mainly found in the membranes of chloroplasts, although clusters of the signal were also found in the interior of these organelles in both types of studied plants. During the analysed period, ACC was not present in the nucleus, while significant amounts were found in the nuclear envelope (Appendix A).

After 1 h and 3 h of hypoxia, the level of the *ACS2* transcripts decreased compared to the control conditions, in wt and *ncb-1* (Appendix A). This phenomenon may be associated with a relatively large amount of ACC in normoxia which is sufficient for ethylene synthesis in the first stages of hypoxia (Appendix A). In the subsequent stages, the level of ACS2 mRNA increased and reached the highest value after 6 h in wt and after 5 days of hypoxia in *ncb-1* (Appendix A). The expression pattern of the *ACS2* gene correlates with the greater amount of ACC after 5 days of hypoxia in *ncb-1* (Appendix A). Next, the level of ACO2 mRNA, which oxidizes ACC to ethylene and is responsible for the final quantity of hormone in cells, was investigated (Appendix A). In all stages of hypoxia, the amount of ACO2 transcripts was higher in wt than in *ncb-1*. After 5 days of oxygen deprivation, the level of ACO2 mRNA in wt was more than twice as high as that in *ncb-1* (Appendix A). This may result in a higher ACC amount in *ncb-1* due to a less efficient conversion to ethylene (Appendix A). It appears that higher levels of ACC and ACS2 mRNA in a mutant at some stages of hypoxia, and higher levels of ACO2 mRNA in wt suggest different ethylene signalling patterns during hypoxia in wt and *ncb-1*.

To determine how plants respond to induced stress, we examined the expression level of the hypoxia response gene *ADH1* (Figure 1). It was revealed that there was an increase in ADH1 mRNA expression in the first hours (3, 6) of stress in wt plants. *ADH1* upregulation in *ncb-1* was not noticed during this period. Conversely, during long hypoxia (5 days), a higher transcript level was found in the mutant. Reoxygenation caused a strong reduction in *ADH1* expression in both wt and *ncb-1*. It thus follows that the mutant reacts to the oxygen deficiency is delayed, which may affect growth conditions.

We previously demonstrated the participation of CBs in the accumulation of poly(A^+^) RNA during hypoxia stress in the roots of *A. thaliana* and *Lupinus luteus* [38,41]. Next, we studied the prevalence of this phenomenon in leaf cells. To investigate whether hypoxia induces a change in poly(A^+^) RNA distribution and accumulation and whether CBs may be involved in this process, U2B”:GFP (one of the marker proteins of Cajal bodies) and *ncb-1* mutants *A. thaliana* were studied. For this purpose, microscopic analysis and the measurements of the amount of poly(A^+^) mRNA in nuclei, CBs, and cytoplasm were performed (Figure 2A–J). Whole-mount FISH was conducted on leaves grown under normoxic, hypoxic, and reoxygenation conditions. The U2B”:GFP signal revealed nuclei in both genotypes and CBs in wt. In the cytoplasm, poly(A^+^) RNA, which can participate in the translation, was present in the form of an extensive network between chloroplasts (Figure 2). Hypoxia caused a significant reduction in the quantity of transcripts in both plants. The strongest decrease in the amount of poly(A^+^) mRNA was observed after 5 days of hypoxia vs. normoxia, 2.6 times in wt and over 5.5 times in *ncb-1* (Figure 2D,I and Figure 3A). For the first time, significant changes were observed under hypoxic stress between wt and *ncb-1*. Significant differences between the two plant types were observed after one day of reoxygenation, and the amount of poly(A^+^) RNA was more than twice as high in wt than in *ncb-1* (Figure 2E,J and Figure 3). In the cytoplasm, the level of poly(A^+^) mRNA was lower by only approximately 30% in wt and as much as over 70% in *ncb-1* compared to normoxia. This indicates that the quantity of mRNA recovers more quickly to physiological levels in wt.

In the nucleus, homogeneous signal of poly(A^+^) RNA was detected in the nucleoplasm, except in the nucleolus (Figure 2A’1,B’1,D’1,I’1). A reduction in the level of poly(A^+^) RNA was observed during oxygen deprivation conditions, similar to the cytoplasm (Figure 3B). However, under long-term submergence (5 days), poly(A^+^) RNA accumulated in the nuclei of wt (Figure 2D). This is 2,7 times more than in *ncb-1* (Figure 3B). The results show RNA retention in the nucleus during deep hypoxic stress in wt plants. After the removal of the stress, the level of poly(A^+^) RNA in the nucleus decreased in wt and increased in *ncb-1* (Figure 2E,J). We previously showed the participation of CBs in the retention of poly(A^+^) RNA in the nucleus in the *A. thaliana* roots [38]. For this reason, we checked the level of transcripts in CBs in wt leaves (Figure 3C). In mesophyll *A. thaliana* cells, regions of the nucleus with strong signals and CBs usually overlap. In contrast to root cells, a precise accumulation of poly(A^+^) RNA in CBs was only occasionally observed in the leaves (Figure 2A’1–A’3). The quantity of poly(A^+^) RNA in CBs slightly decreased during hypoxia, but after 5 days of stress, it was the highest (Figure 3C). In CB-deficient mutants, there are disturbances in nuclear retention and a decrease in the amount of poly(A^+^) RNA in the cytoplasm during prolonged hypoxia and reoxygenation respectively, which may affect the condition of plants.

In the next step, we checked whether the changes in quantity and localization of poly(A^+^) RNA in both genotypes were related to the level of transcription. To this end, immunolocalization and the amount of phosphorylated Ser2 in the CTD domain of RPB1, the largest subunit of RNA polymerase II (the active form of the polymerase) (Figure 4A–F), were analysed. A homogeneous signal with small speckles except for the nucleolus occurred in the nucleus under normoxia, hypoxia, and reoxygenation. In the first stages of hypoxia (6 h), an increase in the amount of active RNA polymerase II in wt and *ncb-1* was observed (Figure 4B,B’). In the following stages, the level of transcription decreased. After 5 days of submerging in water, when poly(A^+^) RNA accumulated in the nucleus, the amount of active polymerase II was still decreasing (Figure 4E,E’). The amount of active RNA polymerase II was 21.8% and 13.2% of the normoxia level in wt and *ncb-1*, respectively. At all stages of hypoxia, the quantity of active RNA polymerase II was lower in *ncb-1* (Figure 4G). Consequently, during reoxygenation, the level of transcription strongly increased in wt (Figure 4F) and only slightly in *ncb-1* (Figure 4F’). The obtained results indicate that the accumulation of poly(A^+^) RNA in the nucleus is not accompanied by a strong increase in the level of transcription (Figure 4G).

Subsequently, we checked whether changes in the level of RNA polymerase II are reflected at the RPB1 (largest subunit of RNA polymerase II) transcription level (AT4G35800) (Figure 5A). Expression measurements of two amplicons of RPB1, including the eighth and ninth exons (E8E9) and the third exon and third intron (E3I3), were performed for wt and *ncb-1* (Figure 5B,D). The results obtained by qPCR showed that the level of RPB1 mRNA correlated with the amount of active RNA polymerase II and quantity of poly(A^+^) RNA during hypoxia. The level of RPB1 expression of E8E9 was almost twice lower after one day of hypoxia in both the *ncb-1* and wt compared to the control conditions (Figure 5B). A further decrease in the quantity of E8E9 transcripts was noted for *ncb-1* after five days of hypoxia. The wt plants at this stage had 35% more transcripts than the mutant. On the other hand, a rapid increase in the level of RPB1 mRNA occurred after a one-day reoxygenation (5D + 1D) for both tested plants. However, the expression level for the E8E9 amplicon was twofold higher for wt than for the *ncb-1* mutant, which directly corresponds to the amount of active RNA polymerase II.

Next, the analysis of alternative splicing isoforms was performed with particular emphasis on intron (RI) retention in RPB1 mRNA using PCR (Figure 5C and Appendix A). RI is the most common splicing event and affects the first introns [55,56]. Only in mutants, there were transcripts with introns 2 and 3 during hypoxia (Appendix A). This was revealed using primers to flank the exons. Then, exon-intron junction primers were used for the first 6 introns (Appendix A). The results confirm the presence of introns 2 and 3 only in *ncb-1* (Appendix A). Additionally, the sequencing of the amplicon generated with primers for exons 3 and 4 in wt and *ncb-1* under control and hypoxic conditions was conducted. It thus follows that only in mutant is an isoform with intron 3 (Appendix A). Subsequently, primers were used for exons 1 and 4, and the results showed that *ncb-1* contains only two isoforms, without or with both introns 2 and 3 (Appendix A). Transcripts with introns occurred in *ncb-1* in the control conditions and five-day hypoxia groups (Figure 5C). The electrophoretic fluorescence ratio of the isoform with introns to the isoform without introns was one and three quarters higher in the five-day hypoxia than in the control (Figure 5C). These data indicate that the occurrence of RI is much more frequent in transcripts under hypoxia than under normoxia. Consequently, the quantity of spliced isoforms is much less vs. isoform with introns (Figure 5C,D). In turn, the analysis to find PTCs (*ang. premature termination codons*) showed two and three PTCs in intron 2 and 3 mRNA RPB1, respectively (Appendix A). mRNAs with PTCs are degraded through nonsense-mediated mRNA decay (NMD) in the cytoplasm. However, relative expression measurements showed that RPB1 mRNA, including introns, showed a large amount of this isoform in the mutant (Figure 5D). Under stress, the level of isoforms with introns was approximately 40% higher than that under control conditions. This indicated that isoforms with RI are not subject to NMD. In turn, an increase in the amount of mRNA with introns in *ncb-1* implies that disruption of the splicing system by removal of CBs promotes intron retention, especially during abiotic stress. In turn, after *ncb-1* reoxygenation, the isoform with introns largely disappears with increasing amounts of fully mature mRNAs (Figure 5B,D).

To determine whether any isoforms mRNA RPB1 is retained in the nucleus and avoiding NMD, we performed FISH with probes for intron 3 and exon 8 in isolated nuclei and on sections of leaves under physiological conditions and during five-day hypoxia (Figure 6A–H). Accumulation of isoforms with intron 3 in the nuclei of *ncb-1* during hypoxia was observed (Figure 6F and Figure 7). In long hypoxia, the quantity of these exceeded the control values and was the highest among all tested variants in the nucleus (Figure 7). The mRNA with intron 3 was poorly visible in wt and the quantity could not be measured (Figure 6A,B,F and Figure 7). Next, the localization of mRNA with exon 8 was performed (Figure 6C,D,G,H). The probes to exon 8 also identify an isoform with intron 3 in *ncb-1*. The amount of transcripts detected with the probe to exon 8 was the same as using the probe to intron 3 in the mutant (Figure 7). After five days of hypoxia, the level of transcripts in nuclei increased more than twofold in *ncb-1* (Figure 7). This result indicates that in the nuclei of *ncb-1*, there is predominantly an isoform with intron 3 in hypoxia stress (Figure 6H). Accumulation in the nucleus of wt, which was at a lower level than in *ncb-1*, was observed after in situ hybridization with the probe to exon 8 in wt. The amount of the signal increased by approximately 50% compared to the control. The obtained results reveal the retention of transcripts in the nucleus in both genotypes; however, two different AS isoforms accumulated, and higher levels were observed in the mutant. In situ hybridization conducted on the leaf sections showed that only a small amount of mRNAs with intron 3, indicated by small dots, are in the cytoplasm of *ncb-1*, which could be associated with ribosomes (Figure 6E’,F’). mRNA with exon 8 on sections was abundant in the cytoplasm in both genotypes under normoxia and hypoxia (Figure 6C’,D’,G’,H’). However, the strongest signal was found in wt under oxygen deprivation conditions.

## 3. Discussion

It has been shown that many genes fine-tune transcription in response to abiotic stresses in plants [56]. Our results for the detection of poly(A^+^) RNA show the overall response of the transcriptome by changing its localization and quantity in leaf cells subjected to hypoxia. We previously demonstrated strong poly(A^+^) RNA retention, including coding transcripts in nuclei and CBs, in roots of *Lupinus luteus* and *A. thaliana* submerged in water [38,41]. The mechanism is probably a component of plants’ protective strategy that is taken up to survive adverse environmental conditions [38]. Therefore, whether such accumulation in nuclei is organ-dependent was studied. In *A. thaliana* mesophyll cells, retention of poly(A^+^) RNA in nuclei was observed with decreasing transcription levels. However, this phenomenon was observed only under prolonged hypoxia in wt. The level of retention of poly(A^+^) RNA in leaves was lower than that in roots [38] and never exceeded the amount observed under normoxia. Currently, RNA retention in the nucleus is considered an element of gene expression regulation, which may be responsible for the degradation or storage of mRNA for subsequent stages of development or recovery after stress [36]. Retention of mRNA in the nucleus was observed in the microspore of *Marsilea vestita* development. Stored RNAs whose translation is essential and is post-transcriptionally regulated after processing and transport to the cytoplasm are expressed at specific times and in specific cells of the gametophyte during spermatogenesis [56]. Nuclear transcriptome analysis also showed an accumulation of certain mRNAs during hypoxia in roots that are highly translated upon reaeration [39]. However, the form of these mRNAs and the mechanism of accumulation in nuclei are unknown. Our results showed that the level of poly(A^+^) RNA in the nuclei of leaves decreases during reoxygenation and increases in the cytoplasm. This finding supports the hypothesis that after stress, retained poly(A^+^) RNA is transferred from the nucleus to the cytoplasm for translation. However, this phenomenon is much less intense than that found in the root cells. Our results reveal that the distribution of poly(A^+^) RNA under stress could be dependent on the metabolism of cells and the amount of RNA in cells and plant organs. If nuclear RNA retention is an important component of the oxygen deprivation condition response, weaker changes in spatial regulation of gene expression in leaves may result from the fact that this is usually the last of the plant organs to submerge during floods and their hypoxia compared to roots in soil is therefore lower.

A stronger accumulation of poly(A^+^) RNA in the nucleus was observed in wt vs. *ncb-1*, indicating a CBs function in this process. In wt, the amount of poly(A^+^) RNA decreased in CBs in mesophyll cells during the first stages of hypoxia. In the plants submerged for 5 days in water, the poly(A^+^) RNA increased compared to 3 days of stress. At the same time, in contrast to root cells, a stronger concentration of poly(A^+^) RNA in CBs vs. surrounding nucleoplasm was observed less frequently. This indicates that the function of CBs in RNA retention in the leaves’ nuclei could be different from that in the roots and may rely on regulating the quantity of transcripts in the nucleus in a way other than only the accumulation/storage of transcripts. Recently, Rudzka et al. (2022) [49] showed the involvement of CBs in the retention of mRNA isoforms with introns in larix microspores. Only non-fully spliced Sm protein transcripts were stored in the CBs. Retained introns are spliced at precisely defined times, and fully mature mRNAs are released into the cytoplasm, where Sm proteins are produced [49]. Our results show a slightly different pattern retention of RNA with introns in the nuclei of somatic cells. We studied the retention of introns in RBP1 in wt and *ncb-1* with a complete absence in CBs. mRNA RPB1 is one retained in the nucleus, in Lee and Bailey-Serres (2019) [39] studies but the splicing status transcript was not revealed. Our results revealed retention of two of the first six introns of mRNA RPB1 in *ncb-1* under control and oxygen-deprived conditions. The transcripts with introns are the majority of the total RPB1 mRNA pool during hypoxia in *ncb-1*. The FISH results with a probe for intron 3 revealed that this isoform is strongly retained in the nucleus. Despite the lack of CBs, isoforms with introns accumulate strongly in nuclei, especially during stress. Additionally, the percentage of isoforms estimated by PCR revealed that more than 80% of the amount detected in nuclei by the probe to exon 8 detects mRNA with introns, so the accumulation of the form without introns is low in *ncb-1*. This result may be due to a lack of CBs and the need to provide RNA polymerase II under stress survival. Our research shows that intron retention in transcripts is more important than CBs for the accumulation of RNA in nuclei. It has been shown in plants [57,58] and in animals [59] that mRNAs containing introns are found in the nucleus, but various nuclear structures may be involved in this process. For example, in animal cells, mRNA with introns are in nuclear speckles where the post-transcriptional process of their removal may take place [60,61]. In plants, non-fully spliced transcripts were detected in nuclear speckles [57] in microspores of *Marsilea vestita* and in CBs in *Larix decidua* [49]. Our results showed that RNA with introns accumulated in the nuclei devoid of CBs in leaf cells. It thus follows that the mechanism and domains related to mRNA retention in the nucleus in leaves may be distinct from those in generative cells. However, our data show that CBs, even if it is unnecessary, strongly promote the accumulation of poly(A^+^) RNA in leaves’ nuclei. This is because we observed a stronger signal of poly(A^+^) RNA in wt vs. *ncb-1*. Under hypoxia, fully mature RPB1 mRNA was retained in the nucleus. Lowering the amount of mRNA in nuclei and CBs with a simultaneous increase in the amount in the cytoplasm during reoxygenation suggests their use in translation. In conclusion, CBs in plants are involved in mRNA retention in the nucleus; however, the strength and mechanism of this phenomenon may depend on the type of cells, plant organ, and isoform of RNA.

It has been shown that mutation of some core spliceosomal components (luc7, gemin2, lsm8, and rbm25) had a strong effect on exon skipping and RI events regulated by abiotic stress conditions [62,63,64]. We show that depletion of the nuclear domain i.e CBs has a similar effect. It is believed that CB proteins are associated with snRNA biogenesis [65,66]; therefore, the lack of sufficient snRNP complexes also impacts AS in plants. The isoforms of RPB1 with both 2 and 3 introns with PTCs were only in *ncb-1*. AS generates PTC-harbouring nonsense isoforms and is frequently referred to as unproductive alternative splicing (UAS), and mRNAs harbouring in-frame PTCs may trigger a nonsense-mediated mRNA decay (NMD) pathway [67]. Almost 20% of all multiexon, protein-encoding genes in *A. thaliana* were found to generate at least one splicing variant that was upregulated upon NMD impairment [68,69]. However, only 1–2% of protein-coding genes are degraded by NMD [70,71]. The RPB1 mRNA with introns also mostly escapes NMD because its amount increases strongly during hypoxia. It cannot be ruled out that a small amount of RPB1 mRNA with introns undergoes NMD because the localization of this isoform showed a weak signal in the cytoplasm where stalled during translation begins to direct the mRNA to degradation. As our research has shown, most mRNAs with introns can escape NMD because they accumulate strongly in the cell nucleus. It follows that improperly spliced mRNAs due to a disturbance of the splicing system are not immediately degraded. Such transcripts can be protected in the nucleus until subsequent signals decide their fate.

The retention of introns in mRNA has been described in a few abiotic stress and model plants [52,56,72,73]. Recently, it was shown that introns are removed from only 28% of such transcripts and can then be transported to the cytoplasm [73]. The mechanism and location of posttranscriptional splicing need clarification. However, it has been shown that mutants PROTEIN ARGININE METHYLTRANSFERASE 5 (PRMT5) and SKI-INTERACTING PROTEIN (SKIP) strongly suppress splicing but retain only introns [73]. PRMT5 is implicated in various developmental processes, such as flowering time control, stress response, and circadian rhythm, by promoting the recruitment of the NineTeen Complex to the spliceosome and modulating pre-mRNA splicing of diverse genes [74,75]. However, the fate of such RNAs is unknown. Our data suggest that isoforms RPB1 with introns are not unproductive RNA because transcripts can be used in reoxygenation. This is indicated by a strong decrease in the amount of mRNA with introns and an increase in fully maturated transcripts after the removal of stress. Consequently, an increase in the amount of RNA polymerase II was observed. It follows that from the accumulated non-fully spliced mRNAs in *ncb-1* nuclei during hypoxia, introns are removed after reoxygenation. Then, the mRNA is transferred to the cytoplasm, and RPB1 is translated. This process enables intense transcription, allowing recovery after stress.

In conclusion, our results indicate that splicing defects resulting from the lack of CBs reduce abiotic stress tolerance. Our findings also indicate the mechanism of this phenomenon. In *ncb-1*, ethylene biogenesis is impaired. There are probably lower levels of hormones in the mutant that regulate the response to hypoxia. It has been shown that mutant ERFs exhibit poorer survival under reduced aerobic conditions [24]. The results of the ADH1 mRNA measurement indicate a later shift to anaerobic respiration *ncb-1*, which causes a greater energy deficit than in wt. There is a stronger reduction than at wt, the level of transcription, the quantity of poly(A^+^) RNA, and their retention in the nuclei. Prolonged stress affects the AS mRNA of RPB1, resulting in a significant increase in the quantity of isoforms with introns and a decrease in the level of fully spliced mRNA. Despite the post-transcriptional splicing of mRNA with introns during reoxygenation, the levels of RNA polymerase II in *ncb-1* were lower than those in wt. This leads to *ncb-1* leaves being more yellow and curly after 5 days of stress but does not cause necrosis; however, after reoxygenation, *ncb-1* are lower and have fewer flower stems.

## 4. Materials and Methods

### 4.1. Plant Growth and Treatment

*Arabidopsis thaliana* ecotype Columbia-0, U2B”’:GFP and *ncb-1* [54] mutants were grown at 22 °C under 150 mM m^2^/s light for 14 days in a long-day growth chamber. Hypoxia was induced by submerging plants in pots in plastic containers filled with tap water (the oxygen concentration in the water: 8.7 mg/L) so that the water level was approximately 5 cm above plants, and the plants were grown in the same conditions but under low light (30–60 µM m^2^/s).

### 4.2. Immunolocalization of ACC (1-Aminocyclopropane-1-carboxylic Acid)

For sample preparation, the small pieces of leaves were fixed in 4% paraformaldehyde (Polysciences, Warrington, PA, USA) and 0.25% glutaraldehyde (Merck, Darmstadt, Germany) in PBS pH 7.2 overnight at 4 °C. The material was dehydrated in increasing ethanol concentrations containing 10 mM dithiothreitol (DTT) (Thermo Fisher Scientific, Waltham, MA, USA), supersaturated and then embedded in BMM resin (butyl methacrylate, methyl methacrylate, 0.5% benzoil ethyl ether (Merck, Darmstadt, Germany) with 10 mM DTT) at −20 °C under UV light for polymerization. The embedded material was cut on Leica UCT ultramicrotome into semithin sections (1.5 µm), which were placed on microscope slides coated with Biobond (British Biocell International, Cardiff, UK). Before performing immunocytochemical reactions, the BMM resin was removed by washing the sections in pure acetone twice for 15 min and then in water and, finally, in PBS pH 7.2.

The sections were incubated with the primary antibody (Agrisera, Vännäs Sweden) 1:50 in 1% BSA overnight at 4 °C, followed by the secondary antibody 1:250 in 1% BSA for 1 h at 35 °C. After incubation, the sections were washed twice with PBS for 5 min, and then the nuclei were stained with Hoechst 33342 (Thermo Fisher Scientific, Waltham, MA, USA) 1:2500 for 10 min. Sections were then rinsed with PBS solution and RNAse-free water for 5 min and dried for approximately 20 min. The control reaction was performed in the same manner, except that the primary antibodies were omitted.

### 4.3. Whole-Mount In Situ Poly(A^+^) RNA Localization

In situ hybridization was performed as previously described in Germain et al. (2010) [76]. Small pieces of leaves were immersed in 1 mL of fixation cocktail (50% fixation buffer consisting of 120 mM NaCl, 7 mM Na_2_HPO_4_ 3 mM NaH_2_PO_4_, 2.7 mM KCl, 0.1% Tween 20, 80 mM EGTA (Merck, Darmstadt, Germany), 5% formaldehyde (Polysciences, Warrington, PA, USA, 10% DMSO and 50% heptane (Merck, Darmstadt, Germany) for 30 min at room temperature. Next, the samples were treated twice for 5 min each in 100% methanol, three times for 5 min each in 100% ethanol and incubated for 30 min in ethanol: xylene (1:1) (Merck, Darmstadt, Germany) with gentle agitation. The samples were then washed in fixation buffer without formaldehyde and fixed in 5% formaldehyde (Polysciences, Warrington, PA, USA for 30 min. Finally, the cells were prehybridized at 30 °C for 30 min and hybridized overnight at 26 °C with 5′ Cy3 T(T)29 at a concentration of 50 pmol/mL.

### 4.4. RNA Isolation, PCR and qPCR

Total RNA from 100 mg *Arabidopsis thaliana* leaves was isolated using the TRIzol reagent (Thermo Fisher Scientific, Waltham, MA, USA) and the Direct-sol RNA MiniPrep kit (Zymo Research, Irvine, CA, USA).

Reverse transcription was performed using the Transcriptor First Strand cDNA Synthesis Kit (Roche, Basel, Switzerland) with oligo d(T)_18_ primer. The samples were incubated in a thermocycler at 25 °C for 10 min, then 55 °C for 30 min, and finally 5 min at 85 °C (C1000 Touch Thermal Cycler (Bio Rad, Hercules, CA, USA).

Real-time PCR was performed using the LightCycler 480 SYBR Green Master kit (Roche, Basel, Switzerland). The primers used in the procedure were designed for the exons and introns of the gene encoding the largest subunit of RNA II RPB1 polymerase for *Arabidopsis thaliana* (AT4G35800) The sequences of primers are shown in Appendix A). The thermal profile was carried out according to default parameters: a pre-incubation step at 95 °C for 10 min (1 cycle), an amplification step (respectively at 95 °C for 10 s, 52 °C for 10 s, 72 °C for 20 s, where the total number of cycles was 45. The number of transcripts, relative to the reference gene *GAPDH* (At1g13440), *ACT2* (AT3G18780), *UBC5* (AT1G63800)) was calculated using the Light Cycler 96 program. In order to determine the PCR efficiencies, standard curves for target and control genes were obtained using a series of cDNA dilutions as a template. In addition to the LC96 system, the LC96 software (Roche Basel, Switzerland) was used to interpret the amplification curves and estimate the level of transcript expression.

Prior to proceeding with the PCR procedure, the concentration of single-stranded DNA (ssDNA) was measured with a Qubit 4 Fluorimeter (Thermo Fisher Scientific, Waltham, MA, USA) using the Qubit TM ssDNA Assay Kit (Thermo Fisher Scientific, Waltham, MA, USA). A PCR kit with Color OptiTaq DNA Polymerase (EURx) was used for the procedure. A mixture was prepared with a final volume of 25 μL according to the manufacturer’s procedure. The PCR reaction was carried out in a C1000 Touch Thermal Cycler (Bio Rad, Hercules, CA, USA) with the following parameters: initial denaturation at 95 °C for 3 min (1 cycle), denaturation at 95 °C for 30 s, annealing at 56 °C for 30 s, extension at 72 °C for 1 min (35 cycles in total), final extension at 72 °C for 5 min (1 cycle) and cool down at 4 °C indefinitely. To evaluate the PCR products, electrophoresis was performed on a 2% agarose gel. Documentation was prepared using the ChemiDocTM (Bio Rad, Hercules, CA, USA) tactile imaging system.

Elution of the DNA fragments from the agarose gel was performed with the Gene MATRIX Agarose-Out DNA Purification Kit (EURx). Sequencing was performed by Genomed S.A. (Warsaw, Poland). The DNA sequences of the individual samples were analysed by software Fast PCR software [77].

### 4.5. Exposure of Nuclei for Microscopic Analysis and Immunolocalization of the Phosphorylated Serine 2 CTD Domain of RNA Polymerase II

For nuclei isolation, about 100 leaves were fixed for 20 min in 4% paraformaldehyde solution (Polysciences, Warrington, PA, USA). Then the material was chopped as finely as possible with a razor blade in the presence of 750 µL of lysis buffer (15 mM Tris-HCl at pH = 7.5, 2 mM NaEDTA, 0.5 mM spermine, 80 mM KCl, 20 mM NaCl, 0.1% Triton X-100). The was diluted by adding 3 mL sorting buffer (100 mM Tris-HCl at pH = 7.5, 50 mM KCl, 2 mM MgCl_2_, 0.05% Tween-20.5% sucrose) to which was added 3% BSA (Merck, Darmstadt, Germany). About 80 μL of the mixtures were then spread onto a glass slide and dried at 35 °C. These samples were used for localization of RNA polymerase II and FISH to mRNA of RPB1.

The slides with isolated nuclei were incubated with rat anti-phosphorylated serine-2 in the CTD domain of RNA POL II (Chromotek; Germany) diluted 1:100 in 1% BSA in PBS pH = 7.2 overnight at 4 °C. Next incubation with goat anti-rat antibodies labelled with Alexa Fluor 546 (Thermo Fisher Scientific, Waltham, MA, USA) for 1 h at 36 °C was performed. The control reaction was performed in the same manner, except that the primary antibodies were omitted.

### 4.6. FISH to mRNA RPB1

For sample preparation, the small pieces of leaves were fixed in 4% paraformaldehyde (Polysciences, Warrington, PA, USA) and 0.25% glutaraldehyde (Merck, Darmstadt, Germany) in PBS pH 7.2 overnight at 4 °C. The material was dehydrated in increasing ethanol concentrations containing 10 mM dithiothreitol (DTT) (Thermo Fisher Scientific, Waltham, MA, USA), supersaturated and then embedded in BMM resin (butyl methacrylate, methyl methacrylate, 0.5% benzoil ethyl ether (Merck, Darmstadt, Germany) with 10 mM DTT) at −20 °C under UV light for polymerization. The embedded material was cut on Leica UCT ultramicrotome into semithin sections (1.5 µm), which were placed on microscope slides coated with Biobond (British Biocell International, Cardiff, UK). Before performing immunocytochemical reactions, the BMM resin was removed by washing the sections in pure acetone twice for 15 min and then in water and, finally, in 4xSSC pH 7.0.

FISH was conducted for a minimum of 16 h after 1 h prehybridisation step using hybridization buffer (50% (*v*/*v*) (Merck, Darmstadt, Germany) with 30% (*v*/*v*) formamide (Merck, Darmstadt, Germany) on isolated nuclei and resin sections. For exon 8 and intron 3 mRNA of RPB1 localisation probe respectively 5′dig-CTGTTT TGGTATGATAAGATTAAAAACTTGTTT3′ and 5′dig-GTAAAATTCATTCGAAAGAAAAGA CGAAC3′. After wash steps in SSC buffer incubation with rabbit anti-digoxygenin antibodies (Thermo Fisher Scientific, Waltham, MA, USA) diluted 1:100 in 1% BSA in PBS pH = 7.2 in 4 °C was performed overnight. Then goat anti-rabbit antibodies with Alexa Fluor 488 diluted 1:500 in 1% BSA in PBS pH = 7.2) were used. DNA was stained for 10 min with Hoechst 33342 (Thermo Fisher Scientific, Waltham, MA, USA) diluted 1:2500 in PBS pH = 7.2. The control reaction was performed in the same manner, except that the primary antibodies were omitted.

### 4.7. Microscopy Analysis, Quantitative Measurement of Fluorescence and Statistical Analysis

A Leica Sp8 confocal scanning microscope equipped with Hc PL APO 63×/1.40 oil objects with 63-fold magnification was used to analyze the results. The photos were recorded using the Leica LAS AF software (Leica, Wetzlar, Germany), Images were acquired sequentially in the blue (Hoechst 33342), green (Alexa Fluor 488, GFP), red (Cy3, Alexa Fluor 546) channels. The optical sections were collected at 0.5-lm intervals. For image processing and analysis, ImageJ (NIH, Bethesda, MD, USA) software was used. The obtained data were corrected for background autofluorescence determined based on negative control signal intensities. All measurements were conducted at the same magnification, area, and time of laser scanning. The analysis of the quantity of poly(A^+^) RNA, RNA polymerase II, and mRNA RPB1 was performed on the basis of 3 independent biological replicates. From 32 to 66 cells or nuclei were analyzed for each antigen and experiments. Statistical analysis was performed using the PAST program. To compare all groups and to determine if there were any significant differences between them, a nonparametric Kruskal–Wallis test was used. To test between which group differences existed, a Mann–Whitney U test with Bonferroni correction was used.

## Figures and Tables

**Figure 1 ijms-23-07568-f001:**
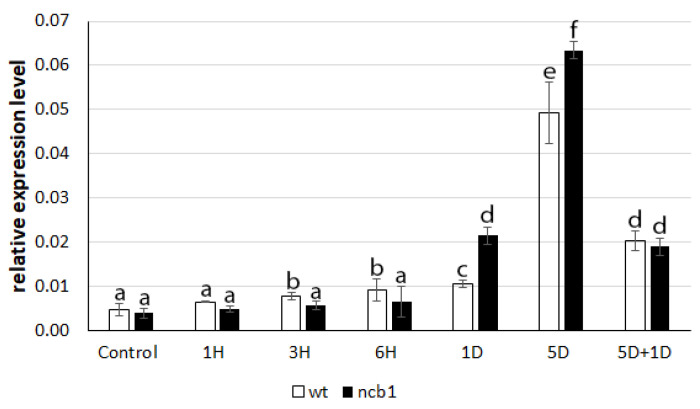
The relative expression levels of mRNA ADH1 (alcohol dehydrogenase 1). X-axis control (Control), time of hypoxia (1, 3, 6H—hours and 1, 5D—days of hypoxia, 5D + 1D—5 days hypoxia and 1-day reoxygenation). Variants labeled with the same letters are not significantly different (*p* ≤ 0.05).

**Figure 2 ijms-23-07568-f002:**
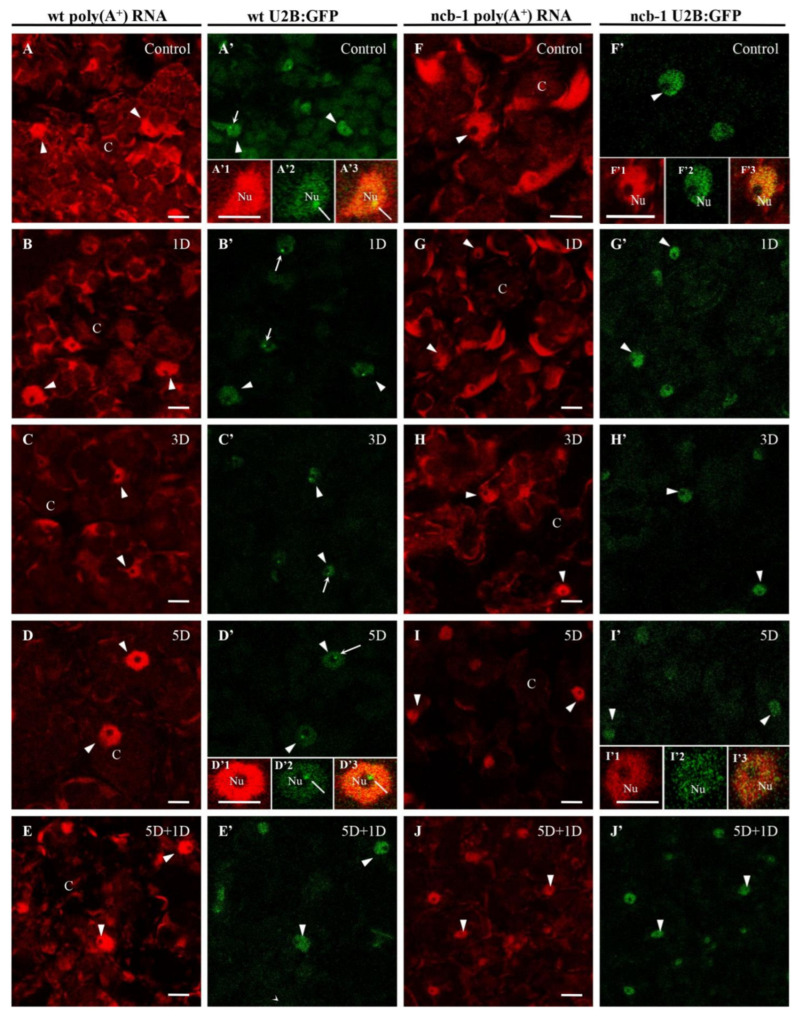
Localization of poly(A^+^) RNA by whole-mount FISH in leaves in wt U2B”:GFP (**A**–**E**) and *ncb-1* U2B”:GFP (**F**–**J**) cultivated to control (**A**,**F**) hypoxia (**B**–**D**,**G**–**I**) and reoxygenation (**E**,**J**) condition. Identification nuclei and CBs by U2B”-GFP (**A’**–**J’**). 1, 3, 5H—hours and 1, 5D—days of hypoxia, 5D + 1D—5 days of hypoxia and 1-day reoxygenation (indicated in the top right corner of the image). The inserts show enlarge, nuclei, and CBs in wt (**A’1**–**A’3**,**D’1**–**D’3**) and *ncb-1* (**F’1**–**F’3**,**I’1**–**I’3**). Arrowhead—nuclei, arrow—Cajal bodies; C—cytoplasm, Nu—nucleolus; in inserts localization of poly(A^+^) RNA w nuclei (**A’1**,**B’1**,**D’1**,**I’1**), identification of nuclei and CBs (**A’2**,**B’2**,**D’2**,**I’2**) with U2B”-GFP and merge (**A’3**,**B’3**,**D’3**,**I’3**). Bar 10 µm.

**Figure 3 ijms-23-07568-f003:**
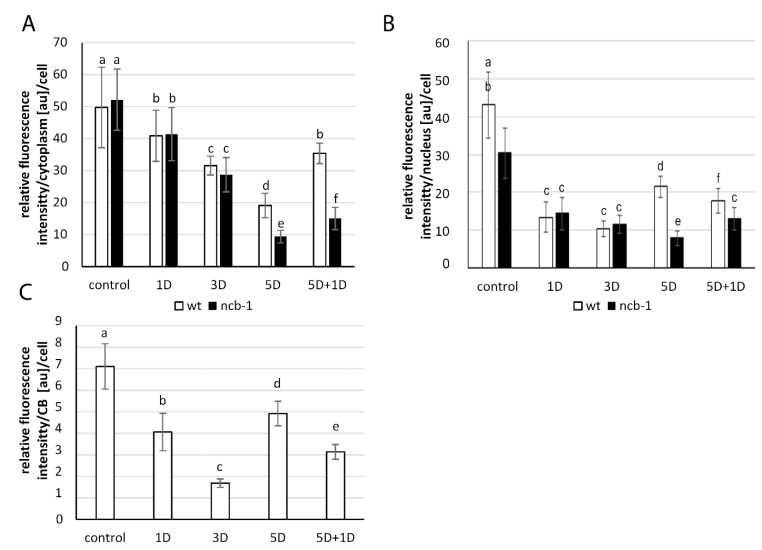
The relative fluorescence intensity of poly(A^+^) RNA in cytoplasm (**A**), nucleus (**B**), CBs (**C**). X-axis—1, 3, 5D—days of hypoxia, 5D + 1D—5 days hypoxia and 1-day reoxygenation. Variants labeled with the same letters are not significantly different (*p* ≤ 0.05).

**Figure 4 ijms-23-07568-f004:**
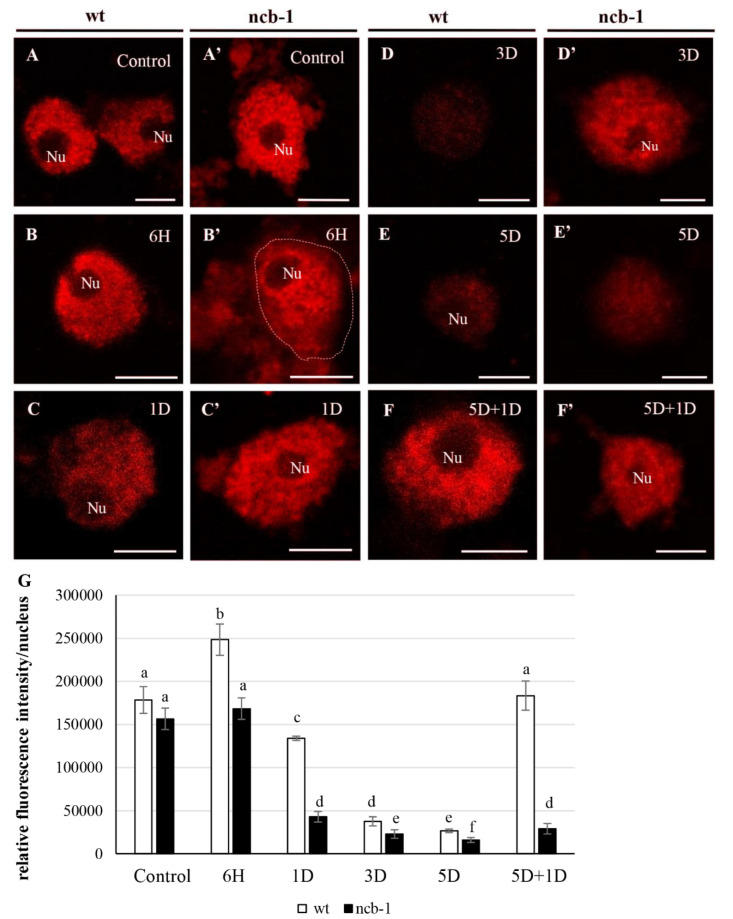
Localization and quantitative analysis of active RNA Pol II in the roots cells *Arabidopsis thaliana.* Distribution of ser2 of CTD RNA Pol II RNA in wt (**A**–**F**) and *ncb-1* (**A’**–**F’**) in control (Control), hypoxia (6H, 1D, 3D, 5D) and reoxygenation condition (5D + 1D) (indicated in the top right corner of the image). In B’ dashes line indicate nucleus. Histogram shows the quantity of ser2 of CTD RNA Pol II RNA in nuclei in different of stages (**G**). X-axis—6H- hours and 1, 3, 5D—days of hypoxia, 5D + 1D—5 days of hypoxia and 1-day reoxygenation. Variants labeled with the same letters are not significantly different (*p* ≤ 0.05). Bar 5 µm.

**Figure 5 ijms-23-07568-f005:**
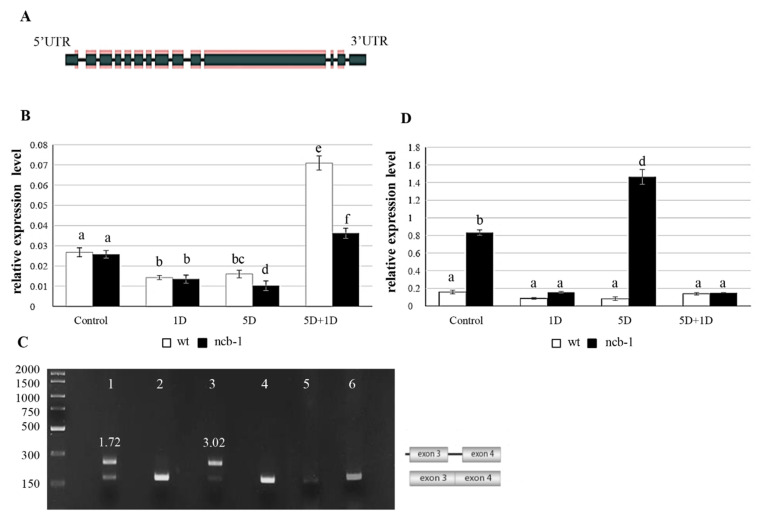
Structure of the largest RPB1 subunit for RNA polymerase II in *Arabidopsis thaliana* (the thicker dark green bars represent exon sequences, and the thinner bars indicate intron sequences) [55] (**A**), Analysis of quantity and ratio of isoforms mRNA RPB1 with intron retention and without introns. The relative expression of transcripts of *RPB1* with exon 8 and 9 (**B**), Gel separation of PCR products with specific primers to exon 3 and 4 mRNA RPB1. line 1—Control *ncb-1*-; line 2—Control wt; line 3—hypoxia *ncb-1*; 4—hypoxia wt; 5—five-day hypoxia and one day reoxidation *ncb-1*; 6—five-day hypoxia and one-day reoxidation wt (**C**), The relative expression of transcripts of *RPB1* with intron 3 in wt and *ncb-1* (**D**). Variants labeled with the same letters are not significantly different (*p* ≤ 0.05). 1D—one-day hypoxia, 5D—five-day hypoxia, 5D + 1D five-day hypoxia and one-day reoxygenation.

**Figure 6 ijms-23-07568-f006:**
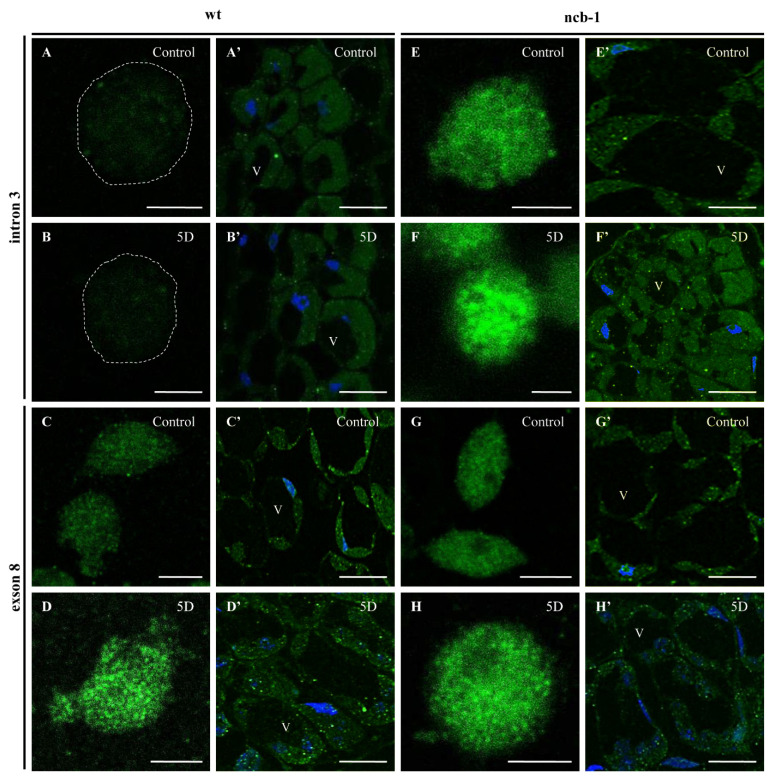
Localization of mRNA RPB1 with intron 3 in the nuclei (**A**,**B**,**E**,**F**) and leaf sections (**A’**,**B’**,**E’**,**F’**) and exon 8 in the nuclei (**C**,**D**,**G**,**H**) and leaves sections (**C’**,**D’**,**G’**,**H’**). In A and B dashes line indicate nucleus. Control—control condition, 5D—five day of hypoxia (indicated in the top right corner of the image). Bar: 5 µm (**A**–**H**) and 100 µm (**A’**–**H’**), V—vacuole. On sections nuclei are stained with DAPI.

**Figure 7 ijms-23-07568-f007:**
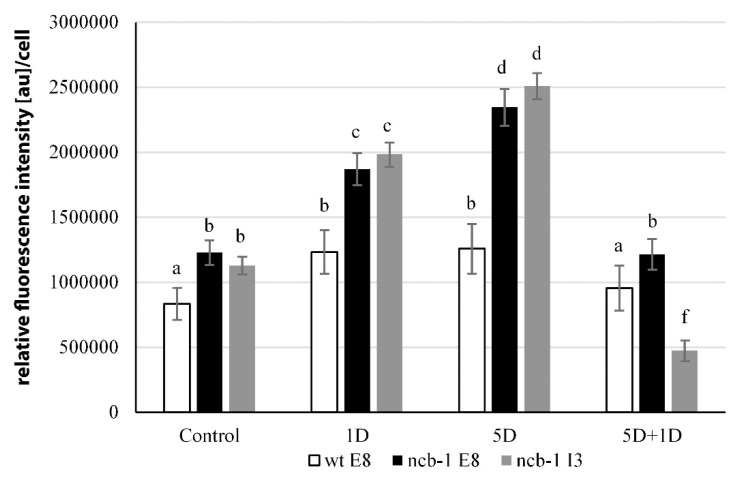
Measurements of the amount of RPB1 mRNA with exon 8 and intron 3 in nuclei wt and *ncb-1* in normoxia (Control), hypoxia (1D, 5D) and reoxygenation (5D + 1D). Variants labeled with the same letters are not significantly different (*p* ≤ 0.05).

## Data Availability

The data and material supporting the findings of this study are available from the corresponding author (D.K. and J.N.) upon request.

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
