# Peer review of "Function of Cajal Bodies in Nuclear RNA Retention in A. thaliana Leaves Subjected to Hypoxia"

_ijms, 2022, doi:10.3390/ijms23147568_

Round 1

Reviewer 1 Report

The authors have made remarkable achievements in the functional study of CB in plants. Their recent study confirmed that the presence of CB in the nucleus plays a significant role in the response of plants to hypoxic stress. Authors also successfully demonstrated that nuclear retention of transcripts plays a positive role in coping with hypoxic stress.

To confirm the specific role of CB in the hypoxic stress environment, they successfully conducted several delicate and careful experiments and drew meaningful results. Eventually, they proved that CB-deficient mutant plants had splicing defects, making them weak against hypoxic stress. They used immunolocalization, immunoquantitation, and various molecular biological methods to prove their hypothesis. They successfully showed clear PCR results or immunostaining of molecular markers such as ethylene biosynthesis-related genes, RPB1, and ADH1. Nevertheless, the authors have unfortunately damaged the paper’s readability by failing to use proper English expressions. Incorrect expression and grammatical errors were found throughout the manuscript, which may be detrimental to the quality. I recommend that the authors read the manuscript carefully once again, correct any errors, and resubmit. In addition, I would like to point out a few things in the text.

1. In lines 427-428, the plants were grown under low light conditions. Even under these conditions, photosynthesis can occur, and oxygen is generated. Please check whether the oxygen concentration inside and outside the plant was measured during the experiment.

2. Typos and errors must be corrected.

Here are just a few examples:

  ex1) Line 434, dithiothreitol (DDT) -> DTT

  ex2) Line 444, Hoest -> Hoechst

  ex3) Lines 491-495, font size and line spacing do not match with the previous sentence

  ex4) Lines 510-511, Alexa 546 -> Alexa Fluor 546

  ex5) In Figure 6 caption, leaves sections -> leaf sections

3. In Materials and Methods section 1.5. In the title, the authors stated “isolation of nuclei.” However, looking at the experimental process, it seems more appropriate to describe it as “exposure of nuclei” rather than “isolation of nuclei”.

4. In Figure 1, the authors should explain how they calculated the relative expression level. It is necessary to explain what was used as the standard value.

5. Converting the large numbers on the Y-axis of Figure 3 to an exponential scale will make it look more concise.

6. Does Figure3-C show only wt data?

7. It is necessary to explain why there is no difference between wt and ncb-1 in hypoxia 1D and 5D+1D in Figure 4-C.

8. For a smooth explanation of several introns and exons of the RBP1 gene used in this study, it seems necessary to include the RBP1 gene structure in Figure S3-A in the figure of the main text.

Author Response

Dear Reviewer

Thank you for the review of our manuscript entitled "Function of Cajal bodies in nuclear RNA retention in A. thaliana leaves subjected to hypoxia", comments helped us to improve the paper. Below are the answers to all comments:

  • Measurements of the amount of oxygen in the water in which the plants were immersed (to stimulate hypoxia) were made.The amount of oxygen in hypoxia only slightly decreased (from 8.7 mg/l in control to 7.9 mg/l in 5 days hypoxia).However, 10 000 times slower diffusion and low solubility of oxygen in water lead to plant hypoxia.In our experiment, the increase in ADH1 mRNA (marker of hypoxia) after 5 days of flooding the plants clearly indicates the of hypoxia.Measuring oxygen inside the plant is very difficult, however, an increase in ADH1 indicates a shift in metabolism to anaerobic. Unfortunately, we do not have information about photosynthesis, however our preliminary data indicate a decrease level of chlorophyll during hypoxia.
  • Typos and errors have been corrected
  • Corrected section header to: Exposure of nuclei for microscopic analysis and immunolocalization of the phosphorylated serine 2 CTD domain of RNA polymerase II
  • The number of transcripts of ADH1 and other mRNA in this manuscript, relative to the reference gene GAPDH (At1g13440), ACT2 (AT3G18780), UBC5 (AT1G63800) was calculated using the Light Cycler 96 program. In order to determine the PCR efficiencies, standard curves for target and control genes were obtained using a series of cDNA dilutions as a template. This information is in 1.4. RNA isolation, PCR and qPCR (Materials and methods)
  • Since the values are relative fluorescence measurements, the scale is reduced.
  • 3C shows the amount of poly (A) RNA in CB in wt, while in ncb1 has no CBs
  • I thank you for this suggestion especially because it turned out that we got the pictures wrong between wt and ncb1 for the 1D stage.However, in my opinion, in the figure for the 5D + 1 stage, the difference in the amount between wt and ncb1 is clearly visible. Maybe it does not fully reflect the difference resulting from the diagram, but not every photo of the measured nuclei with low antigen levels can be shown in the figure (the outline of the nucleus edge is not visibleor the signal is not homogeneous).
  • The diagram of the gene structure has been added to Fig. 5 of the main text.

Reviewer 2 Report

Authors have identified the role of Cajal bodies in the maintenance of nuclear RNA in leaves of A. thaliana that have been exposed to hypoxia. Overall, the manuscript is well written and therefore it can be accepted for publication in the present after a careful English language and spelling check..

Author Response

Dear Reviewer,

Thank you for the review of our manuscript entitled "Function of Cajal bodies in nuclear RNA retention in A. thaliana leaves subjected to hypoxia", comments helped us to improve the paper. The article was additionally revised by a native speaker.